# Prognostic Value of Neoadjuvant Chemotherapy in Patients with Borderline Resectable Pancreatic Carcinoma Followed by Pancreatectomy with Portal Vein Resection and Reconstruction with Venous Allograft

**DOI:** 10.3390/jcm11247380

**Published:** 2022-12-12

**Authors:** Jin-Can Huang, Bing Pan, Han-Xuan Wang, Qing Chen, Qiang He, Shao-Cheng Lyu

**Affiliations:** Department of Hepatobiliary Surgery, Beijing Chao-Yang Hospital Capital Medical University, Beijing 100020, China

**Keywords:** borderline resectable pancreatic cancer, portal vein resection and reconstruction, venous allograft, neo-adjuvant chemotherapy, CA19-9

## Abstract

Background: Neo-adjuvant chemotherapy (NAC) represents one of the current research hotspots in the field of pancreatic ductal adenocarcinoma (PDAC). The aim of this study is to evaluate the prognostic value of NAC in patients with borderline resectable pancreatic cancer (BRPC) followed by pancreatectomy with portal vein (PV) resection and reconstruction with venous allograft (VAG). Methods: Medical records of patients with BPRC who underwent pancreatectomy with concomitant PV resection and reconstruction with VAG between April 2013 and March 2021 were analyzed retrospectively. Outcomes of patients with and without NAC (NAC, Group 1 vs. non-NAC, Group 2) were compared with focus on R0 resection rates, morbidity, and survival. Results: Of the 77 patients with pancreatectomy, PV resection and reconstruction with VAG were identified. Overall survival (OS) rates of 0.5-, 1-, and 2-year were 80.5%, 59.7%, and 31.2%, respectively (median survival time, MST, 14 months). Of these, 24 patients (Group 1) underwent operation following received NAC, and the remaining 53 patients did not (Group 2). The R0 resection rate of vascular margin was 100% vs. 84.9% (*p* = 0.04), respectively. Morbidity of post-operative pancreatic fistula (POPF) was 0% vs. 17.8% (*p* = 0.07), respectively. The OS of 0.5-, 1- and 2-year and MST of 2 groups were 83.3%, 66.7%, 41.7%, 16 months, and 79.2%, 55.6%, 26.4%, 13 months, respectively. Multivariate analysis revealed that carbohydrate antigen 19-9 (CA19-9) serum level and postoperative chemotherapy were independent prognostic factors in patients with BRPC after surgery. Conclusion: NAC might improve the R0 resection rate and POPF in patients with BRPC who underwent pancreatectomy with concomitant PV resection and reconstruction with VAG. Survival benefit exists in patients with BRPC who received NAC before pancreatectomy. Postoperative chemotherapy also had a favorable effect on OS of BRPC patients. Elevated CA 19-9 serum level is associated with poor prognosis, even after NAC-combining operation.

## 1. Introduction

Due to the high tumor malignancy, susceptibility to local vascular invasion and distant metastases [1], pancreatic ductal adenocarcinoma (PDAC) has been on the rise over the last decade, becoming one of the most common causes of cancer death, with a 5-year OS rate of approximately 9% [2]. It was previously believed that although considerable progress had been made in the treatment of PDAC, surgery remains the only potential treatment for patients with PDAC. Borderline resectable pancreatic carcinoma (BPRC) refers to PDAC with vascular invasion that can be resected and reconstructed safely [3]. Few of these patients are eligible for radical surgery, and the higher probability of recurrence leads to a 2-year OS rate of only 30–42% in these operable patients [4].

At present, pancreatectomy combined with venous resection and reconstruction for BRPC has been widely acknowledged [5,6], but the specific method of reconstruction is still controversial. Combined with years of experience in performing pancreatectomy, our center innovatively proposes the staging of vascular invasion in PDAC patients and corresponding surgical management strategies for various staging types [7,8]. On this basis, our center selectively applies venous allograft (VAG) for the resection and reconstruction of veins invaded by PDAC tumor.

In recent years, multimodality treatment including radical surgical resection and chemotherapy has been considered by most guidelines as the gold standard for the treatment of BRPC [9,10,11]. However, a huge proportion of patients have still not received any adjuvant therapy. It has been reported that 40–53% of patients could not receive any postoperative antitumor therapy, owing to the progression of the disease itself, postoperative complications of surgery, and cachexia [12]. Therefore, neoadjuvant chemotherapy (NAC) is considered as a possible treatment option for patients with BRPC [13]. As reported previously, NAC could improve the tumor stage, R0 resection rate, and postoperative mortality rate [14,15].

The aim of this study is to evaluate the impact of NAC on the perioperative and survival prognosis of patients with BRPC who underwent pancreatectomy with PV resection and reconstruction with VAG.

## 2. Materials and Methods

### 2.1. Patient Selection

Between April 2013 and March 2021, patients with BRPC who underwent pancreatectomy with PV resection and reconstruction with VAG were included in this study. The criteria for BRPC were defined by the preoperative resectable status of the National Comprehensive Cancer Network (NCCN) guidelines Version 2.2021 [9]. All patients had a confirmed pathological diagnosis by evaluation of the resected specimens. Patients were divided into two groups based on whether they received NAC before the operation. The NAC regimens included: fluorouracil, leucovorin, irinotecan, and oxaliplatin (FOLFIRINOX) or gemcitabine plus nanoparticle albumin-bound (nab)–paclitaxel (GA). All patients provided written informed consent for the surgical procedures, pathological examination, and neoadjuvant or adjuvant chemotherapy regimens. This study was approved for clinical application by the Ethics Committee (Ethics Nr. 2020-D-302) and the Clinical Application Management Committee of Medical Technology of our hospital.

### 2.2. Surgical Technique

The procedures for pancreatic surgery consisted of pancreaticoduodenectomy (PD) with standard lymphadenectomy or total pancreatectomy (TP), as deemed appropriate. The type of PV resection and reconstruction depended on the site and extent of tumor invasion of the vein (Figure 1). In general, the vein on either side of the tumor-involved segment was dissected free and reconstructed with VAG. Perioperative use of heparin was administered on a routine basis. Iliac veins removed during multi-organ harvesting procedures by the transplantation unit were used as grafts. Immediately after harvesting, grafts were stored in University of Wisconsin solution at 4 °C and matched to recipients according to the ABO-system. All anastomoses were performed free of tension with running 6-0 polypropylene sutures. In order to avoid any anastomotic stenosis, the anastomosis was expanded before complete revascularization by releasing the distal clamp first.

### 2.3. Postoperative Management and Surveillance

Subcutaneous daily injection of nadroparin calcium was started on postoperative day 1 (POD1) and ended on POD7; then, lifelong oral aspirin was recommended. Doppler ultrasound of the reconstructed vein was performed routinely on POD3 and POD7. CT scan combined with angiography was performed to evaluate the condition of the reconstructed vein one month after the operation—if necessary. Patients were discharged to the local hospital or home as soon as the postoperative course was without suspicion of adverse events.

### 2.4. Data Collection

Demographic information and preoperative data, especially the NAC option, were collected. Intraoperative data, postoperative complications including POPF [16], postoperative hemorrhage [17], delayed gastric emptying (DGE) [18], wound infection, severe diarrhea and PV thrombosis, and survival data were also recorded. OS time was calculated from time of surgery to last follow-up or death. For patients who died, survival time after surgery and cause of death were recorded. For the survivors, postoperative survival time and recurrence status were recorded.

### 2.5. Statistical Analysis

Continuous variables conformed to the normal distribution were expressed as mean with standard deviation (SD), and the non-normal distribution was expressed as median with interquartile range (IQR). The Chi-square test or Fisher’s exact test was used for categorical variables, and rank sum test for quantitative variables. The Kaplan–Meier method was used to estimate the overall survival, and the univariate log-rank test was used for the comparison of overall survival between the two groups; *p* < 0.05 was considered statistically significant. All analyses were performed using the SPSS version 19.0, for Microsoft Windows.

## 3. Results

In the study, a total of 77 patients underwent PD (*n* = 61) or TP (*n* = 16) with concomitant PV resection and reconstruction with VAG. Patients were divided into two groups based on whether they received NAC before the operation.

### 3.1. Perioperative Details between Two Groups

Of the 77 patients, 24 patients (Group 1) underwent the operation following NAC, the remaining 53 patients (Group 2) directly received operation without following NAC. A total of 18 patients received GA regimens, and 6 patients received FOLFIRINOX regimens. In Group 1, 8 patients underwent TP.

Perioperative patient characteristics are summarized in Table 1. There was no major difference in the distribution of age (*p* = 0.75), gender (*p* = 0.76), preoperative leucocytes (*p* = 0.11), hemoglobin (*p* = 0.59), platelets (*p* = 0.45), albumin (*p* = 0.33), total bilirubin (*p* = 0.42), CA19-9 (*p* = 0.90), operating time (*p* = 0.24), tumor differentiation (*p* = 0.24), tumor size (*p* = 0.18), lymph node metastasis (*p* = 0.62), postoperative complications (*p* = 0.12), or hospital-stay (*p* = 0.37) among the two groups. The R0 rate of the vascular margin of Group 1 is higher than Group 2 (100% vs. 84.9%, *p* = 0.04) (Figure 2, Appendix A).

The perioperative indexes in patients who underwent PD were compared between the two groups as listed in Table 2. The POPF rate in Group 1 was lower than in Group 2 (0% vs. 17.8%, *p* = 0.07), although there was no statistical difference (Figure 2).

### 3.2. Survival and Prognostic Factors

For all 77 patients, the DFA rate and OS rate of 0.5-, 1-, and 2-year were 72.7%, 54.5%, 18.1%, and 80.5%, 59.7%, and 31.2%, respectively (Figure 3A,B). The overall DFS time and MST were 12 and 14 months. The DFS rates of 0.5-, 1- and 2-year of the two groups were 83.3%, 66.7%, 29.2%, and 68.0%, 49.1%, and 13.2%, respectively (*p* = 0.05, Figure 4A). The OS rates of 0.5-, 1- and 2-year were 83.3%, 66.7%, 41.7%, and 79.2%, 55.6%, and 26.4%, respectively (*p* = 0.04, Figure 4B). The overall median DFS time and MST of Group 1 and Group 2 were 14, 16, and 11, 13 months, respectively.

To clarify useful prognostic factors in patients with BRPC who underwent operation, univariate and multivariate survival analyses were performed for this cohort. Univariate analysis demonstrated that NAC (*p* = 0.04), preoperative CA19-9 (*p* < 0.001), intraoperative blood loss (*p* = 0.04), tumor differentiation (*p* = 0.04), and lymph node metastasis (*p* = 0.04) were significantly associated with overall survival (Table 3). Serum CA19-9 level and postoperative chemotherapy were found to be independent prognostic factors by multivariate analysis (*p* = 0.01, *p* = 0.02) (Table 4). Patient had a better prognosis when preoperative serum CA19-9 level was lower than 400 U/mL or patients received postoperative chemotherapy [19].

## 4. Discussion

Despite new advances and breakthroughs in improving anti-tumor treatment, surgical outcomes, and the OS rates of patients with PDAC are rarely satisfactory. As a relatively rare type of PDAC compared to resectable or unresectable, BRPC is a newly proposed concept, whose definition and standardized treatment options currently remain debatable. In 2006, Varadhachary et al. [20] proposed criteria for BRPC for the first time, defining BRCP patients for surgical treatment. Afterwards, the American Hepato-Pancreato-Biliary Association issued an expert consensus about BRPC in 2009 [21]. Since then, extended radical pancreaticoduodenectomy combining with revascularization began to dominate in the treatment of BRPC. With the application of NAC in pancreatic cancer, especially the FOLFIRINOX regimens, more and more retrospective studies reported the improvement in R0 resection rate and long-term prognosis in BRPC patients after NAC-combining surgery [22,23]. Since 2014, NCCN guidelines have advocated for BRPC patients with high-risk factors (high CA19-9, large primary tumor, enlarged lymph nodes, significant weight loss, and severe pain) to receive NAC, apart from direct surgery.

In 2016, Matthew et al. [24] published the results of the phase III A021101 study on the FOLFIRINOX regimens for BRPC, which laid the foundation for NAC as a first-line treatment option for BRPC. Therefore, NCCN guidelines recommended NAC as a priority for all patients with BRPC, including FOLFIRINOX or GA [25]. A prospective foreign study included 223 patients with BRPC given FOLFIRINOX for 3–6 months, in which the MST of these patients reached 19.2 months. A total of 34 cases underwent operation with a R0 resection rate of 62.3%, and the OS was significantly prolonged compared with 162 unresected patients (30.0 vs. 16.5 months, respectively) [26]. However, controversy about NAC and direct surgery does exist, mainly focusing on the surgical conversion rate of NAC, the standardized course of treatment, and optimal timing of surgery [27]. Most of the current literatures with high-quality evidence comes from Europe and the United States, and the comparison of prognosis is often limited to patients with successful NAC conversion versus direct surgery [28], lacking an overall comparison of the two treatment groups. Therefore, the real outcome after NAC in the Chinese population is still unknown. As the univariate analysis suggested in this research, NAC might be a possible factor affecting the prognosis of patients with BRPC who underwent allogeneic vascular replacement. However, the assumption was not further confirmed in the multivariate analysis. From Conroy’s research [29]. PDAC patients receiving FOLFIRINOX could get better prognosis compared with receiving gemcitabine. In our study, most patients received gemcitabine-based regimens because of poor physical conditions, or some patients failed to get a comprehensive therapeutic cycle before the operation. Considering the relatively small sample size of this study, the potential impact of NAC on prognosis still needs to be verified in a large sample.

For the management of intraoperative portal venous system invasion in patients with BRPC, the international study group on pancreatic surgery classifies them into four types: direct sutures or venous patches can be chosen for local resection, and reconstructions including end-end sutures or vascular replacement can be adopted in segmental resection. However, it does not clearly identify the specific criteria for choosing the type of reconstruction. Vascular reconstruction using vascular replacement materials is often recommended for patients with longer vascular resection lengths (>4 cm) to avoid hidden problems such as high vessel tension resulting from end-to-end anastomosis. Our center proposes that the advantage of using vascular replacement material is that revascularization is not limited by the length of vein resection. For BRPC patients with concomitant portal venous system invasion, this advantage could expand the surgical indications and improve R0 resection rate of the vascular margin [30]. The postoperative pathological results showed that the R0 resection rate of vascular dissection was further improved in patients who received NAC. This is a direct advantage of NAC itself for inhibiting the progression of tumor micrometastases and primary foci, inducing tumor cells into a dormant state, killing free tumor cells and potential micrometastases, and further reducing the chance of vascular invasion, decreasing the risk of postoperative recurrence or metastasis [31]. We believe that the significantly increasing R0 resection rate of pancreatic dissection could be related to the neoadjuvant treatment, which promotes fibrosis of the tumor periphery and reduces the occurrence of microvascular and nerve metastases. This conjecture still needs to be confirmed by a larger sample study.

In addition, regarding the controversial question of whether NAC reduces the incidence of pancreatic fistula, Hank [32] conducted a research including 753 patients, which showed that the rate of clinically relevant POPF was 3.6-fold lower in patients receiving neoadjuvant therapy compared with these receiving upfront resection (3.8% vs. 13.8%, *p* < 0.001). Mungroo et al. [33] analyzed the currently accepted international risk score for POPF and revealed the potential advantage of NAC in reducing POPF. Considering the number of variables affecting POPF, analysis of a large sample study is needed to find a final conclusion.

CA19-9 is still an important serologic marker for clinical diagnosis and prediction of prognosis in PDAC patients [34]. A decreasing level of CA9-9 could reflect the effect of treatment to some extent, and is associated with a good prognosis [35,36]. However, it is difficult to identify which kind of patients require preemptive NAC. Some scholars have suggested that NAC should be prioritized for patients with high CA19-9 level [37]. In our current analysis, when assessing the risk factors associated with survival, even after receiving NAC, initial high CA19-9 level (>400 U/mL) remained an independent risk factor for the overall survival of patients. We consider that this may be mainly related to biliary obstruction of most patients in our data, which resulted in impaired biliary excretion and overall relatively high CA19-9 concentration.

The main limitation of this study is the small number of patients in each group, which might limit the accuracy of our assessment. Second, it represents the experience of our single center only. Future studies, preferably with larger patient cohorts from multi-centers, are needed to further confirm our preliminary outcomes.

## 5. Conclusions

NAC might improve the R0 resection rate and POPF in patients with BRPC who undergo pancreatectomy with concomitant PV resection and reconstruction with VAG. The survival benefit exists in patients with BRPC who underwent NAC before pancreatectomy. Elevated CA 19-9 serum levels are associated with a poor prognosis, even after NAC-combining surgery. Postoperative chemotherapy could also have a good impact on the OS of BRPC patients. We hope that future studies will delve into how to screen patients for effective NAC, so that it may be possible to truly individualize treatment for patients and ultimately improve their overall prognosis.

## Figures and Tables

**Figure 1 jcm-11-07380-f001:**
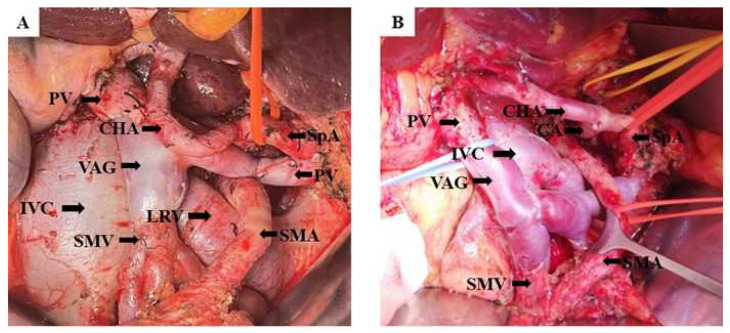
Intraoperative pictures of the portal vein reconstruction. (**A**) Intraoperative picture of patients in Group 1; (**B**) Intraoperative picture of the patients in Group 2. Abbreviations: CA, celiac axis. CHA, common hepatic artery. IVC, inferior vena cava. LRV, left renal vein. PV, portal vein. SMV, superior mesenteric vein. SMA, superior mesenteric artery. SpA, splenic artery. VAG, venous allograft.

**Figure 2 jcm-11-07380-f002:**
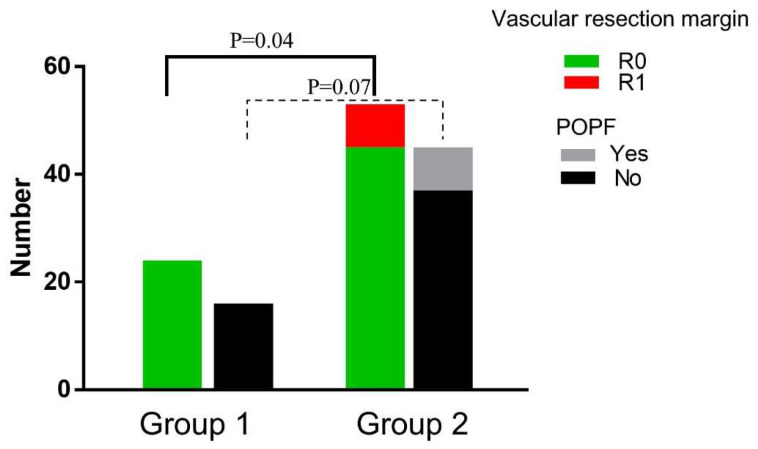
Perioperative data compared between the groups. The vascular resection margin was compared between Group 1 and Group 2 receiving PD and TP; POPF was compared between Group 1 and Group 2 receiving PD. Abbreviations: POPF, postoperative pancreatic fistula. PD, pancreaticoduodenectomy. TP, total pancreatectomy.

**Figure 3 jcm-11-07380-f003:**
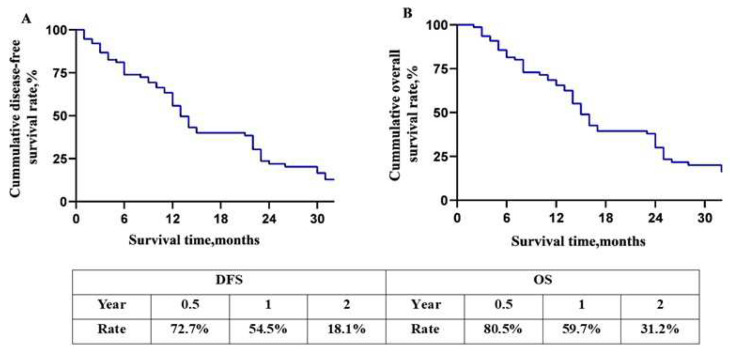
Overall long-term prognosis of the patients. (**A**) Disease-free survival curve of patients with BRPC; (**B**) overall survival curve of patients with BRPC. Abbreviations: DFS, disease free survival. OS, overall survival.

**Figure 4 jcm-11-07380-f004:**
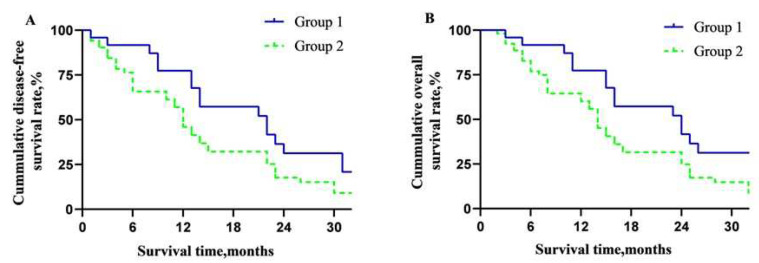
Overall long-term prognosis of the patients in the two groups. (**A**) Disease-free survival curve between the two groups; (**B**) overall survival curve between the two groups.

**Table 1 jcm-11-07380-t001:** Comparison of the general information between the two groups.

Parameters	Group 1 (*n* = 24)	Group 2 (*n* = 53)	*p*-Value
Gender (Male/Female)	10/14	24/29	0.76
Age (Year) [mean ± SD]	62.5 ± 9.9	60.6 ± 11.0	0.75
Preoperative leucocytes(×10^9^/L) [mean ± SD]	4.4 ± 1.5	6.9 ± 3.0	0.11
Preoperative hemoglobin (g/L) [mean ± SD]	120.0 ± 16.2	119.9 ± 17.9	0.59
Preoperative platelets (×10^9^/L) [mean ± SD]	200.6 ± 77.9	213.4 ± 80.1	0.45
Preoperative albumin (g/L) [mean ± SD]	37.8 ± 6.5	38.1 ± 4.4	0.33
Preoperative total bilirubin (μmol/L) [mean ± SD]	67.0 ± 86.5	82.2 ± 89.6	0.42
Preoperative CA19-9 (U/mL) [mean ± SD]	989.6 ± 1807.3	1039.9 ± 1777.8	0.90
Preoperative drainage for jaundice (Yes/No)	3/21	13/40	0.36
History of diabetes (Yes/No)	6/18	16/37	0.84
Intraoperative blood loss, (mL) [median (IQR)]	500 (200, 2600)	600 (200, 2200)	0.41
Intraoperative blood transfusion (Yes/No)	12/12	27/26	0.93
Operation time (h) [mean ± SD]	13.5 ± 2.6	12.4 ± 2.7	0.24
Tumor size (cm) [mean ± SD]	3.8 ± 2.03	3.7 ± 1.3	0.18
Tumor differentiation (low/medium-high)	10/14	15/38	0.24
Pancreatic resection margin (R0/R1)	24/0	46/7	0.06
Vascular resection margin (R0/R1)	24/0	45/8	0.04
Lymph node metastasis (Yes/No)	14/10	34/19	0.62
Postoperative complications (Yes/No)	10/14	13/40	0.12
Postoperative hospital stay (d) [median (IQR)]	28 (14, 106)	20.5 (12, 70)	0.37
Postoperative chemotherapy (Yes/No)	7/17	17/36	>0.99

SD, standard deviation; IQR, interquartile range.

**Table 2 jcm-11-07380-t002:** Comparison of the general information between patients with PD of the two groups.

Parameters	Group 1 (*n* = 16)	Group 2 (*n* = 45)	*p*-Value
Intraoperative blood loss (mL) [median (IQR)]	500 (400, 725)	600 (500, 1000)	0.06
Intraoperative blood transfusion (Yes, %)	10 (62.5)	21 (46.7)	0.27
Operation time (h) [median (IQR)]	12.5 (11.25, 15)	12 (10, 14)	0.21
Pancreatic resection margin (R0, %)	16 (100)	44 (97.8)	0.49
Pancreatic fistula (Yes, %)	0 (0)	8 (17.8)	0.07
Postoperative hospital stays (d) [median (IQR)]	22.5 (16.5, 35.5)	19 (14.5, 28.5)	0.26

**Table 3 jcm-11-07380-t003:** Univariate analysis of postoperative long-term survival in patients with BRPC.

Parameters	Number (*n* = 77)	1-Year Survival Rate (%)	2-Year Survival Rate (%)	*χ* ^2^	*p*-Value
Gender				0.024	0.87
Male	34	67.9	40.2		
Female	43	69.0	36.1		
Age (Year)				0.003	0.95
≤60	30	79.5	45.9		
>60	47	61.2	32.5		
Diabetes				1.770	0.18
Yes	22	57.2	31.2		
No	55	72.9	40.6		
Neoadjuvant chemotherapy				4.110	0.04
Yes	24	77.4	52.1		
No	53	64.5	31.6		
Drainage for jaundice				0.913	0.33
Yes	16	68.8	31.3		
No	61	68.5	40.1		
Albumin (g/L)				0.201	0.65
≤35	23	65.2	33.5		
>35	54	70.1	40.0		
Total bilirubin (U/L)				0.034	0.85
≤60	42	65.6	43.9		
>60	35	71.3	32.7		
Preoperative CA19-9 (U/mL)				14.171	<0.001
≤400	43	87.8	52.7		
>400	34	42.4	16.3		
Operating time (h)				0.522	0.47
≤10	18	59.8	35.9		
>10	59	71.1	38.2		
Intraoperative blood loss (mL)				3.922	0.04
≤500	37	74.9	50.3		
>500	40	68.1	25.5		
Blood transfusion				2.675	0.10
Yes	39	63.5	24.4		
No	38	72.8	49.7		
Tumor differentiation				4.019	0.04
Low	25	55.7	23.6		
Medium–High	52	74.0	43.8		
Tumor size (cm)				0.018	0.89
≤2	10	60.0	30.0		
>2	67	69.9	39.2		
Pancreatic resection margin				1.492	0.22
R0	70	68.5	39.9		
R1	7	68.6	17.1		
Lymph node metastasis				4.047	0.04
Yes	48	64.8	26.9		
No	29	74.7	57.8		
Postoperative chemotherapy				3.141	0.07
Yes	24	87.1	52.2		
No	53	60.0	31.1		

**Table 4 jcm-11-07380-t004:** Multivariate analysis of long-term survival in patients with BRPC.

Factors	RR Value	95% CI	*p*-Value
Neoadjuvant chemotherapy	1.662	0.909–3.039	0.09
Preoperative CA19-9	2.031	1.171–3.523	0.01
Intraoperative blood loss	1.423	0.828–2.446	0.20
Tumor differentiation	1.726	0.956–3.116	0.07
Lymph node metastasis	1.710	0.960–3.045	0.06
Postoperative chemotherapy	2.070	1.135–3.774	0.02

## Data Availability

The data used and analyzed in this study is included in the article or are available from the corresponding and first authors upon reasonable request.

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
