# Peer review of "Prognostic Value of Neoadjuvant Chemotherapy in Patients with Borderline Resectable Pancreatic Carcinoma Followed by Pancreatectomy with Portal Vein Resection and Reconstruction with Venous Allograft"

_jcm, 2022, doi:10.3390/jcm11247380_

Round 1

Reviewer 1 Report

Dear Authors,

You present a very interesting patient population with portal venous resection and reconstruction (venous allograft) in which you studied the impact of neoadjuvant chemotherapy. 

I think the study is of general interest for publication in JCM. However, there are still significant shortcomings that require a thorough revision of the manuscript. 

First, the entire manuscript needs language revision. Sentences are clearly too long and often difficult to understand. 

A relevant bias is that patients with total pancreatectomy (TP; n = 16/77; 20.8%) were included in the study, although the distribution between groups remains unclear. Tables and figures need to be revised (see below). 

Enclosed is a brief list of comments that came to my attention during review of the manuscript:

I would suggest the following title:

Prognostic value of neoadjuvant chemotherapy in patients with borderline resectable pancreatic carcinoma followed by pancreatectomy with portal vein resection and reconstruction with venous allograft

Abstract

13-16: Neo-adjuvant chemotherapy (NAC) represents one of the current research hotspots in the field of pancreatic ductal adenocarcinoma (PCaPDAC). The aim of this study is to evaluate the prognostic value of NAC in patients with borderline resectable pancreatic cancer (BRPC) followed by and followingpancreatectomy with portal vein (PV) resection and reconstruction with venous allograft (VAG).

17-19: Medical records of patients Patients with BPRC who underwent pancreatectomy with concomitant PV resection and reconstruction with VAG between April 2013 and March 2021 were analyzed retrospectively. and compared between Outcomes of patients with and without NAC (NAC, group 1 vs. non-NAC, Group 2) were compared with focus on R0 resection rates, morbidity and survival.

19-21: Results: Of the 77 patients with pancreatectomy, PV resection and reconstruction with VAG were identified. Of these, 24 patients (Group 1) underwent pancreatectomy with concomitant PV resection and reconstruction in VAG following received NAC, and the remaining 53 patients did not (Group 2). directly received operation without following NAC.

The results in the abstract are given in too much detail. Percentages should also be used instead of numbers (e.g., R0 resection, morbidity, and POPF).

It must be stated in Material and Methods which chemotherapeutic agents were given neoadjuvantly.

Results: Please present all key results with precise data and their precisions, instead of vague wording like “no significant differences”.

Please kindly revise the P value in the tables and the main text:

-If the P<0.001, report "P<0.001".

-If the P value is between 0.001 and 0.01 and less than 0.01, report the specific P value to 3 decimal places, e.g., "P=0.001" "P=0.009".

-If the P≥ 0.01, report the specific P-value to 2 decimal places, e.g. "P=0.01" "P=0.06" "P=0.10" "P=0.90".

-If the P-value is greater than 0.99, report "P > 0.99".

-Do not round P-values, do not report "not significant" simply because the data are greater than an arbitrary value, and do not report only vague bounds such as P<0.05, as described above, but report the exact P-value.

- Line 131 and 133: P values should only be given up to the third decimal place. 

-P value in the tables should be consistent with that in the results section

Among the 77 patients, 16 patients (= 20.8 %) obviously had total pancreatectomy (TP). What was the distribution of patients with TP between the groups? No POPF rates can be compared without knowing how many patients/group received TP in each case. Patients with TP cannot be compared with pancreaticoduodenectomy (DP) patients in terms of other parameters either (different surgery times, increased risk of complications such as "gastric venous congestion", etc.). Here, a complete recalculation without patients with TP must be performed!

Tables: the units of all parameters (column 1) must be specified differently, e.g.: "preoperative platelets (x109/L) [mean ± standard deviation] Please adapt this accordingly for all parameters

Table 1, gender: please write "male / female" or "men / women", don't use "male / women"

Table 2: it is clearer to give the data in absolute numbers and percentages, e.g.: "lymph node metastases, n [%] 14 [58.3] vs. 34 [64.2]"

The impact of postoperative chemotherapy on survival should be studied multivariately (possible bias!)

Figures 1 and 2: anatomical structures on figures should be labeled (IVC = inferior vena cava; SMV = superior mesenteric vein; CA = celiac trunk; HA = hepatic artery; SA = splenic artery; SMA = superior mesenteric artery; and so on...). This is the same for the histopathological images (Figure 2). Which staining? Which magnification? "Adventitia" is misspelled (A, Figure 2)

Figures: abbreviations should be explained separately for each figure (e.g., OS = overall survival)

Line 128: “24 patients received adjuvant chemotherapy postoperatively” - adjuvant chemotherapy is always postoperatively, isn´t it?

Line 146-147: the same tense should always be used: "Patients had a better prognosis..." please revise for the whole text

Line 163: please don´t use “Professor”, just write “In 2016, Matthew et al. published…”

Lines 186-192: this sentence is much too long and thus hard to follow. Please use short and straightforward sentences. 

I would recommend the abbreviation PDAC (pancreatic ductal adenocarcinoma) instead of PCa because PDAC is better established in the literature

Author Response

Reviewer #1:

  1. The results in the abstract are given in too much detail. Percentages should also be used instead of numbers (e.g., R0 resection, morbidity, and POPF).

Reply 1: Thank you for your advice. We have simplified the results in the abstract and used percentages as suggested.

  1. It must be stated in Material and Methods which chemotherapeutic agents were given neoadjuvantly.

Reply 2: This comment is of great importance to us. We have added the NAC regimens received by patients of our study in the manuscript.

  1. Results: Please present all key results with precise data and their precisions, instead of vague wording like “no significant differences”.

Reply 3: Thank you for your advice. Manuscript was corrected.

  1. Among the 77 patients, 16 patients (= 20.8 %) obviously had total pancreatectomy (TP). What was the distribution of patients with TP between the groups? No POPF rates can be compared without knowing how many patients/group received TP in each case. Patients with TP cannot be compared with pancreaticoduodenectomy (DP) patients in terms of other parameters either (different surgery times, increased risk of complications such as "gastric venous congestion", etc.). Here, a complete recalculation without patients with TP must be performed!

Reply 4: This comment is of great importance to us. We have compared some general information between patients with PD of two groups, and submitted as Table 2 in our manuscript.

  1. Tables: the units of all parameters (column 1) must be specified differently, e.g.: "preoperative platelets (x109/L) [mean ± standard deviation] Please adapt this accordingly for all parameters.

Reply 5: Thank you for your advice. We have revised all the parameters in tables as required.

  1. The impact of postoperative chemotherapy on survival should be studied multivariately (possible bias!)

Reply 6: This comment is of great importance to us. We have studied postoperative chemotherapy in multivariate analysis, finding its significant on patients’ OS.

  1. Figures 1 and 2: anatomical structures on figures should be labeled (IVC = inferior vena cava; SMV = superior mesenteric vein; CA = celiac trunk; HA = hepatic artery; SA = splenic artery; SMA = superior mesenteric artery; and so on...). This is the same for the histopathological images (Figure 2). Which staining? Which magnification? "Adventitia" is misspelled (A, Figure 2)

Reply 7: This comment is of great importance to us. We have labeled anatomical structures on Figures 1 and put original Figure.2A and B in supplementary material. Thank you for your comment!

  1. Line 146-147: the same tense should always be used: "Patients had a better prognosis..." please revise for the whole text.

Reply 8: Thank you for your advice. We have revised the tenses for the whole text.

  1. Line 163: please don´t use “Professor”, just write “In 2016, Matthew et al. published…”

Reply 9: Thank you for your advice. We have revised the manuscript as suggested.

  1. Lines 186-192: this sentence is much too long and thus hard to follow. Please use short and straightforward sentences. 

Reply 10: Thank you for your advice. We have simplified some of the sentences as suggested.

  1. I would recommend the abbreviation PDAC (pancreatic ductal adenocarcinoma) instead of PCa because PDAC is better established in the literature.

Reply 11: Thank you for your advice. We have replaced PCa with PDAC in the manuscript.

Reviewer 2 Report

This paper looked into the effect of neoadjuvant chemotherapy on borderline resectable pancreatic cancer treatment(BRPC) in patients following portal vein resection. There are a few issues that needs to be addressed before considering for publication.

1.       Line 53: The authors indicated that neoadjuvant chemotherapy (NAC) is considered as a possible treatment option for patients with BRPC but there’s no reference provided. Please give more details on the mechanism of NAC to patients in the previous study and indicate the reference. 

2.       For what the authors are talking about in Fig.2, it will be clearer to show the vascular margin and incidence of POPF separately with Bar graph and P value between the two groups of patients and leave the non-significant data in Table 2.

3.       Fig.2A and B, the label on the figure is obscure. Please use a better resolution on it.

4.       In Line 146, the authors indicated that the CA19-9 level below 400U/ml has a better prognosis. Please include references for this. Besides, The CA19-9 level in Table 2 indicates that the preoperative CA19-9 level is 989 in Group 1 and 1053 in Group 2 without no significant difference which is much higher than 400U/ml. Is CA19-9 going to drop after surgery with/without Neoadjuvant chemotherapy? Please provide data for that and explain better.

5.       Based on the comparison between the 2 groups with/without NAC in the BRPC after vein reception, it is significant, but the difference of OS is not really drastic (about 10% difference for each time point), which indicated that NAC might not be that effective on these patients. And it might not be solved by increasing patient numbers if the difference is so small. Please discuss the possible reasons for that which might give some hints for future directions.

Author Response

Reviewer #2:

  1. Line 53: The authors indicated that neoadjuvant chemotherapy (NAC) is considered as a possible treatment option for patients with BRPC but there’s no reference provided. Please give more details on the mechanism of NAC to patients in the previous study and indicate the reference .

Reply 1: Thank you for your comment. We have listed some of the researches concerning NAC on BRPC patients and further explained the mechanism of NAC as required. Your comment is of great importance to our research and make our research more accurate.

  1. For what the authors are talking about in Fig.2, it will be clearer to show the vascular margin and incidence of POPF separately with Bar graph and P value between the two groups of patients and leave the non-significant data in Table 2.

Reply 2: Thank you for your advice. We have made a Bar graph of vascular margin and POPF as required. Original figure was in supplementary material.

  1. 2A and B, the label on the figure is obscure. Please use a better resolution on it.

Reply 3: This comment is of great importance to us. We have updated original Figure.2A and B for a better resolution, and put in supplementary material. Thank you for your comment!

  1. In Line 146, the authors indicated that the CA19-9 level below 400U/ml has a better prognosis. Please include references for this. Besides, The CA19-9 level in Table 2 indicates that the preoperative CA19-9 level is 989 in Group 1 and 1053 in Group 2 without no significant difference which is much higher than 400U/ml. Is CA19-9 going to drop after surgery with/without Neoadjuvant chemotherapy? Please provide data for that and explain better.

Reply 4: Thank you for comment. We referred a relevant article, which confirms our opinion that CA19-9 level below 400U/ml has a better prognosis. We have collected the serum CA19-9 on 1month,3 months and 6 months after surgery and made a Line Char named as “supplementary figure 2”. It can be seen from the figure that CA19-9 significantly decreased after operation, as a result of the tumor resection. However, CA19-9 increase gradually over time after operation, it might have a connection with tumor recurrence. As we have pointed out in the manuscript from line 244 to 246, the much higher level of CA19-9 in our study might be the result of biliary obstruction. Thank you for your comment!

  1. Based on the comparison between the 2 groups with/without NAC in the BRPC after vein reception, it is significant, but the difference of OS is not really drastic (about 10% difference for each time point), which indicated that NAC might not be that effective on these patients. And it might not be solved by increasing patient numbers if the difference is so small. Please discuss the possible reasons for that which might give some hints for future directions.

Reply 5: Thank you for your comment. In our research, the neo-adjuvant chemotherapy regimens were not all the same. Some patients received FOLFIRINOX, others received gemcitabine-based regimens. Firstly, the body conditions of patients in China are relatively poorer, which means most of them cannot endure FOLFIRINOX, and just received gemcitabine-based regimens as the first choice suggested by oncologist. Secondly, gemcitabine-based regimens have a smaller survival advantage compared with FOLFIRINOX. Finally, part of patients who hadn’t gotten recommended cycles of NAC may have a poor OS. Based on these reasons, more patients should be enrolled in the further study and discuss the effect of different NAC regimens of BRPC patients who underwent pancreatectomy with PV resection and reconstruction with venous allograft. Your comment is of great significance to our research!

Round 2

Reviewer 1 Report

Dear Authors,

You have revised the manuscript very thoroughly. Particularly of importance is the additional presentation of all patients with PD. From my side a publication is recommended.

One remark:

The perioperative indexes in patients who underwent PD were compare between two groups as listed in Table 2. POPF rate in Group1 was lower than in Group2 (0% vs.17.8%, P=0.07), although there was no statistical difference (Figure 2). 

--> I think you meant "table 2", not "figure 2", please check!

Author Response

Dear Reviewer,

Thanks for your comments.

Following another reviewer's suggestion, we made a bar graph to show the vascular margin and incidence of POPF in Figure 2.

Should I delete it or not?

Qiang He

Reviewer 2 Report

The authors addressed all my concerns in my first review and I think it is good for publication now. 

Author Response

Thank you very much for your review and comments.
